# An Overview of Phytosanitary Irradiation Requirements for Australian Pests of Quarantine Concern

Humayra Akter [1,*], Nancy Cunningham [1], Polychronis Rempoulakis [2] and Martin Bluml [3]

1   South Australian Research and Development Institute, Department of Primary Industries and Regions SA, Urrbrae, SA 5000, Australia
2   NSW Department of Primary Industries, Ourimbah, NSW 2258, Australia
3   Agriculture Victoria Research, Department of Energy, Environment and Climate Action, Bundoora, VIC 3083, Australia
*   Correspondence: humayra_05@hotmail.com

**Abstract:** Phytosanitary irradiation is used to prevent the introduction or spread of unwanted plant pests and diseases found in horticulture commodities, both in a domestic and international trade setting. Australia started exporting irradiated horticulture commodities to New Zealand in 2004. Since then, exports of irradiated products have continued to grow as phytosanitary irradiation has become more widely accepted for the treatment of plant pests by our international trading partners. Domestically, Food Standards Australia New Zealand (FSANZ) now allows irradiation of all fresh fruits and vegetables using an irradiation dose of 150 to 1000 Gy for all insect pests. To facilitate further domestic and international trade in Australian irradiated horticulture products, we conducted a literature review to perform the following: (1) identify information gaps (minimum absorbed irradiation dose) for Australian pests of quarantine concern, and (2) identify where differences may exist between the minimum absorbed dose and the regulated dose set, and that is accepted by Australia and key international trading partners. In Australia, a minimum absorbed dose of 400 Gy can be used to treat all insect pests of quarantine concern. However, a lower minimum absorbed dose of 150 Gy is used for many fruit fly species that are important for domestic and international trade. For a limited number of priority insect and non-insect pests highlighted by the horticulture sector, there were gaps found for minimum absorbed irradiation dose in the literature. These pests include Vineyard snail, Serpentine leaf miner and Fuller's rose weevil. Studies to establish the minimum absorbed dose for Vineyard snails, Serpentine leaf miners and Fuller's rose weevil are recommended. In addition to the gaps identified for irradiation dose, there is merit in conducting further research to refine (lower) the minimum absorbed dose for specific pests and priority commodities where irradiation has an impact on quality. A reduction in dose may not only benefit product quality but will also reduce both treatment time and cost.

**Keywords:** phytosanitary treatment; irradiation dose; insect; pests

## 1. Introduction

For domestic and international trade purposes, specific protocols must be agreed upon between governments on how pests of quarantine concern will be treated. There is a range of phytosanitary treatment options that can be employed to prevent the introduction or spread of unwanted pests. They protect horticultural produce and help the horticultural sector manage plant biosecurity risks, safeguarding both overseas and domestic trade and ensuring imports are pest free.

Traditional phytosanitary measures include heat treatments (with hot air or hot water at 43–48 °C), cold disinfestation (0–3 °C), chemical dip treatment (usually with dimethoate solutions) and chemical fumigation (with ethylene dibromide and methyl bromide). Phytosanitary measures need to be carried out in a consistent and effective manner and reach the required efficacy for every application and pest species treated. This ensures that

Australian treatment standards align with the principles and obligations of the sanitary and phytosanitary (SPS) agreement governed by the World Trade Organization (WTO). In Australia, The Commonwealth Department of Agriculture, Fisheries and Forestry (DAFF) plays an important role in maintaining and improving technical market access, working closely with international and state jurisdictions to resolve SPS issues and meet Australia's obligations under the WTO SPS agreement [1]. National plant protection organizations (NPPOs) are responsible for specifying the minimum absorbed dose within a phytosanitary protocol and require large-scale confirmatory experiments to provide certainty about dose efficacy.

Concern for health and environmental issues is on the increase worldwide. How farmers grow food and treat the land on which they grow is of increasing importance to consumers. Several of Australia's trade partners have called into question the use of certain phytosanitary treatments and the use of agrichemicals to control pests and diseases of quarantine significance. For example, methyl bromide is known to deplete ozone once released into the atmosphere and is toxic to humans and fauna if not used correctly. In response to this, Australia and many of its trading partners have established strict requirements (protocols) that govern the use of phytosanitary treatments and the accepted level of chemical residues. This has led Australia's many horticultural industries to consider the use of irradiation as a preferred phytosanitary treatment.

Regulatory restrictions and specific pest and commodity requirements imposed by importing countries can inhibit trade; therefore, traditional phytosanitary treatments need to take this into consideration. For instance, New Zealand does not accept heat treatments and fumigation as a disinfestation method for Australian pests. For Australia to meet New Zealand quarantine requirements, alternative phytosanitary treatments such as irradiation are required. Movement of fresh produce between Australian states is also strongly regulated, and like New Zealand, heat treatments and fumigation are not approved for all horticultural products.

Heat treatment adversely affects stone fruits, pome fruits and avocados. Cold treatments can also affect fruit quality through chilling injury, and the process commonly requires a defined period at set temperatures (1–2 °C). In practice, cold treatment for some horticultural commodities (e.g., apple, pear, citrus, etc.) is cost-effective and achievable for both domestic and export markets. Interstate certification assurance protocols (ICAs) ensure that the correct procedures have been taken to meet interstate plant quarantine requirements. For international trade, the Manual of Importing Country Requirements (MICoR) lists the requirements of importing countries for a full range of crops. In the case of more perishable commodities (e.g., apricots, peaches, nectarines, leafy vegetables), extending the period between harvest and retail will usually result in reduced quality, shelf life and price.

In a separate but related literature review of phytosanitary irradiation pathways and product quality tolerance, phytosanitary irradiation was considered comparable and, in some instances, superior to other phytosanitary treatments, albeit with caveats on further research into commodities not studied and compared (Personal communication with John Golding, New South Wales Department of Primary Industries, NSW, Australia). Phytosanitary irradiation, therefore, offers a viable alternative for eliminating these concerns. It is fast and residue-free, with the added advantage that it can be applied to pre-packaged fresh produce [2]. It is effective for a wide range of arthropod pests [3] and particularly effective on internal pests such as fruit flies, and insects have not developed resistance to its application [4]. Consuming irradiated food, including fresh fruit and vegetables, has been shown to be safe for human health and the environment [5,6]. Irradiation is increasingly being seen as an important strategy for improving food hygiene and food safety [3].

Irradiation as a horticultural market access treatment can ensure that no viable insect pest is transported to importing countries [7]. Phytosanitary irradiation is considered acceptable if the following conditions are met:

- The pest dies [8–10];
- The pest is prevented from successful development (no adult emergence) [8,11];
- The pest cannot reproduce (induced sterility), or the treatment inactivates the pest completely [7,8].

Phytosanitary measures are subject to internationally recognized protocols [12], to which Australia is a signatory. A framework of rules for the conduct of trade in irradiated fresh produce was not available until 2003, with the adoption of International Standards for Phytosanitary Measures (ISPM) 18 by the International Plant Protection Commission (IPPC). In following years, ISPM 28 and its annexes specified the minimum absorbed dose for many pests of quarantine concern. Domestically, Food Standards Australia New Zealand (FSANZ) now allows irradiation of all fresh fruits and vegetables at doses between 150 to 1000 Gy for all insect pests.

Irradiation as a phytosanitary treatment, like fumigation with methyl bromide, has a unique advantage in that it works as a broad-spectrum treatment for almost all important regulated arthropod pests [13]. Phytosanitary irradiation protocols are designed to prevent the reproduction of harmful pests through death, the prevention of adult emergence, and adult or F1 generation sterility [14]. Phytosanitary irradiation measures for the imported product as well as domestic and exported products, align with Australian treatment application standards [15].

Australia started exporting irradiated horticulture commodities to New Zealand in 2004. Since then, exports of irradiated products have continued to grow as phytosanitary irradiation has become more widely accepted for the treatment of pests of quarantine concern by international trading partners. There is a significant opportunity to expand the use of phytosanitary irradiation to treat specific pests for priority markets and commodities (Table 1).

## 2. Materials and Methods

For this review, information on the minimum absorbed dose contained in the scientific literature and regulated dose imposed by Australia and key international trading partners were collected for Australian pests of quarantine concern.

Information was sourced through the published peer-reviewed literature using keyword searches in Google Scholar, PubMed and journal websites. The International Database on Insect Disinfestation and Sterilization (IDIDAS), which includes data on dose required for the phytosanitary irradiation (disinfestation) and dose required to induce sterility (sterilization) in target pests, was consulted. Personal communication with experts from Commonwealth and State agencies, irradiation service providers (Steritech, Victoria, Australia) and international experts (USDA) were used to acquire the grey literature, including government and industry reports, websites and standards published by IPPC-FAO and international trading partners.

## 3. Results and Discussion

### 3.1. International Standards for Acceptance of Irradiation Dose for Phytosanitation

The Codex Alimentarius Commission (the Codex General Standard for Irradiated Foods, CXS 106- 1983, Rev.1-2003) [16] and the International Plant Protection Convention (IPPC) international standards for the use of irradiation as a phytosanitary measure [17] provide an international framework and guidelines for plant protection. These standards are recognized by Food and Agriculture Organization (FAO), World Health Organization (WHO), the International Consultative Group on Food Irradiation (ICGFI), and World Trade Organization (WTO). These standards were prepared and endorsed to achieve international harmonization of phytosanitary measures, with the aim of facilitating global trade and avoiding the use of unjustifiable phytosanitary measures as barriers to trade. The IPPC sets internationally recognized protocols and standards for food irradiation, including the International Standard for Phytosanitary Measures ISPM 18 [17]—Guidelines for the use of

irradiation as a phytosanitary measure and ISPM 28 [18]—Phytosanitary treatments for regulated pests, with Part 7 being specific to fruit flies.

A maximum absorbed dose of 1000 Gy to irradiate fresh fruits and vegetables was considered safe for human consumption by the FAO/IAEA/WHO expert committee in 1980, being formally approved in 1983 [19]. A maximum absorbed dose of 1000 Gy has also been approved by the FDA [20] and FSANZ [21]. Thus, keeping irradiation dose within this limit is a prerequisite to facilitating the international trade of irradiated fresh horticultural products.

A minimum absorbed dose of 400 Gy is sufficient to satisfy quarantine regulations without adversely affecting the physicochemical and nutritional value of most fruits and vegetables [22]. In 2006, the USDA-APHIS recommended that a minimum absorbed dose of 400 Gy be used to control broad taxonomic groups of insect pests without affecting the quality of a wide range of commodities [12,23,24]; however, this is yet to be endorsed through an ISPM at international level. Nevertheless, a minimum dose of 150 Gy is generally recognized as an effective treatment for all tephritid fruit flies [9,25] and 400 Gy as a treatment for all remaining pest insects in host fruits and vegetables [25]. A dose of 70 Gy is effective in preventing the emergence of adults of fruit flies of the genus Anastrepha from fruits and vegetables, which is included as PT 39 in ISPM 28 [18]. A notable exception is adult lepidoptera that pupate internally [9,25]. A dose of 250 Gy is sufficient to prevent the emergence of viable adults from eggs and larvae of Tortricidae, which is stated in PT 40 of ISPM 28 [18].

### 3.2. Domestic Standards for Acceptance of Phytosanitary Irradiation

In conjunction with international standards, radiation protection and nuclear safety requirements are regulated in Australia by the Australian Radiation Protection and Nuclear Safety Authority (ARPANSA), while Food Standards Australia and New Zealand (FSANZ) approve the use of irradiation of food. In developing food regulatory measures, FSANZ plays a vital role in promoting consistency between domestic and international food standards in Australia and New Zealand

On May 2021, FSANZ announced a new domestic food regulatory measure that permits the use of irradiation as a phytosanitary treatment for all fresh fruit and vegetables. Although all types of fruits and vegetables are currently approved for irradiation in Australia, only 0.3–8% of total fruit and vegetables consumed in Australia and New Zealand (NZ) are irradiated [21]. Nevertheless, demand for irradiation services is increasing in Australia as more trading partners accept irradiation as a phytosanitary treatment. Regarded as simple and highly reliable, phytosanitary irradiation is opening doors for Australian exporters in Asia. Both Vietnam and Thailand recognize irradiation as a phytosanitary treatment for Australian horticulture products.

According to FSANZ, irradiation must not be used as a substitute for good hygiene, manufacturing and agricultural practices and must be within 150 Gy to 1000 Gy [21]. At present, Australia uses irradiation as a phytosanitary treatment for fresh commodities, including mangoes, cherries, grapes, citrus, lychee, melons and tomatoes, with proven technical and commercial benefits. Commodities such as berries, summer fruit, apples, pears, pomegranate, asparagus, and kiwi fruit are also irradiated before export but in limited numbers due to higher perishability. A list of exportable fresh commodities is attached in Table 1.

The minimum absorbed dose is accepted for Australian domestic trade for the following insect pests/groups; 150 Gy for tephritid fruit flies; 300 Gy for mango seed weevil; and 400 Gy for all other insect pests excluding Lepidopterans that pupate internally (Benjamin Reilly, Fresh Produce Business Manager, Steritech, Victoria, Australia, personal communication).

However, 250 Gy has been applied successfully for the control of arthropods on mango and papaya [13], and 350 Gy for all arthropods on lychee [26] when exported from Australia to New Zealand.

### 3.3. Phytosanitary Irradiation Dose and Efficacy Data for Pests of Quarantine Concern

To ensure the minimum absorbed dose is reached in a commercial irradiation facility, the actual absorbed dose is always higher than the regulated dose imposed by Australia and its trading partners. This is due to the need for a high dose uniformity ratio (DUR) which is commonly between 1.5 to 2.2 for produce treatments in Australia (Personal communication with Benjamin Reilly, Fresh Produce Business Manager, Steritech, Victoria, Australia.) and also could be higher like 2.5 or 3.1 in other countries [27,28]. For example, a dose of 600 Gy to 1000 Gy is delivered to ensure a minimum absorbed dose of 400 Gy is achieved [27,28], while 300 Gy could be delivered to ensure a minimum absorbed dose of 150 Gy is achieved [8].

#### 3.3.1. Tephritids—Fruit Flies

A dose of 150 Gy is accepted for all Tephritid flies [9,25], although, at lower doses, sterilization of some species can ensure effective treatment. For instance, at 75 Gy, no adult *B. tryoni* emerged from a total of 24,700 third instars in orange and avocado [29], and over one-half million third instar *B. tryoni* in apple, orange, avocado, mango, tomato and cherry were irradiated with 75 Gy, and none survived to the adult stage [30]; 138,635 larvae were treated in mango from which no adult Qfly emerged at 74 to 101 Gy [31]. Therefore, for Qfly, 100 Gy is sufficient to inhibit development. Bustos et al. [32] treated 100,000 third instars of each of three species of *Anastrepha ludens*, *Anastrepha obliqua* and *Anastrepha serpentina*, in mangoes with 100 Gy without any adults developing. Thus, 100 Gy is considered effective as a phytosanitary treatment for these species.

#### 3.3.2. Lepidopterans—Moths and Borers

After fruit flies, Lepidopterans, especially tortricid moths, are the most significant pests of economic and quarantine concern, especially species that significantly impact mangoes and summer fruit [33,34]. Lepidopterans are sterilized at a lower dose than 400 Gy. Tortricid moths can be controlled at doses between 120 and 200 Gy [35]. A dose of 200 Gy was shown to be sufficient to control the codling moth, *Cydia pomonella* (L.) [36]; citrus fruit borer *Ecdytolopha aurantiana* (Lima) [37]; oriental fruit moth, *Grapholita molesta* (Busck) [38]; and light brown apple moth, *Epiphyas postvittana* (Walker) [39]. Irradiation tests with litchi borer/moth *Cryptophlebia illepida* (Butler) indicated that 250 Gy prevented adult emergence from irradiated fourth and fifth instars, but 125 Gy was sufficient to cause sterility [40]. Information on several other tortricid species is available from studies undertaken for sterile insect release programs [33–35,41]. These studies concur that 200 Gy is sufficient to sterilize females. Thus, a dose of 200 Gy could be effective as a minimum absorbed dose for tortricids. Nevertheless, large-scale validation tests on several additional species in the same family are desirable before making a full recommendation [42].

Identifying a dose for tortricid moths below 400 Gy could reduce treatment time, thereby reducing the cost of treatment and increasing the volume of products that can be treated at an irradiation facility [12]. For a limited number of fresh products, a lower dose will help minimize or avoid the impact of irradiation treatment on product quality [43].

#### 3.3.3. Thrips

Thrips often result in remedial fumigation in NZ, but studies in Japan showed that a generic treatment dose of 400 Gy [44] could control the pests. Four species of thrips (*Frankliniella occidentalis*, *Frankliniella schultzei*, *Thrips tabaci* and *Thrips imaginis*) exposed to gamma irradiation at doses of 200 Gy became sterile. This would be sufficient to control these species of thrips in fresh produce [45]. A minimum absorbed dose of 250 Gy is recommended as a safety margin until large-scale confirmatory trials are conducted [45]. However, the FAO continues to recommend 150–350 Gy for sterilizing thrips [11].

### 3.3.4. Mites

Australia and New Zealand accept a minimum absorbed dose of 400 Gy for all mites, including Tetranychidae (Acari mite family) [46]. The minimum absorbed dose of 400 Gy is also required by Japan for Acari mites [47]. More specifically, the lychee erinose mite *Aceria litchii* (Keifer) (Trombidiformes: Eriophyidae) is reported to require a dose of 400 Gy to treat [48].

Although the 400 Gy dose is not yet supported by large-scale confirmatory testing, it is high enough to be considered safe [49]. Hallman (2012) [26] also suggested that a recommended dose for acari mites might be as low as 350 Gy. Other studies have shown that a sterilizing dose for Tetranychidae is 300 Gy [50–52]. Additional large-scale species-specific confirmatory testing will likely be required to demonstrate that a lower dose is effective for mites.

### 3.4. Gaps in Dose and Efficacy Data for Australian Pests of Quarantine Concern

There are a limited number of pests impacting horticulture commodities where no minimum absorbed dose information has been found in the literature. These pests of concern include Fuller's rose weevil, Serpentine leaf miner and Vineyard snail.

### 3.4.1. Fuller's Rose Weevil

Fuller's rose weevil (FRW), *Asynonychus cervinus,* has proven to be a major pest of quarantine significance for the citrus industry. FRW lays eggs under the calyx of citrus fruit and is the most common stage of FRW reported on export citrus. The South Australian Research and Development Institute (SARDI) has run several projects to control this pest, focusing on monitoring techniques, egg removal from fruit, biological control and chemical control [53]. Although the removal of eggs from under the calyx has had partial success with various treatments, mortality and removal have not reached an acceptable level and cannot be considered a satisfactory phytosanitary treatment. Management practices are limited to a systems approach, including orchard control through tree skirting, weed control and six weekly applications of chemical spray [54], followed by inline postharvest treatments to reduce FRW incidence from under the calyx.

Golding and Palou [55] and McDonald et al. [56] examined the effect of low-dose irradiation on fruit quality in citrus, but less research has been undertaken to determine an effective minimum absorbed dose, and no work has been performed on phytosanitary irradiation for FRW in Australia. Studies in the USA [57,58] found that FRW eggs failed to hatch at irradiation treatment levels of ≥50 Gy. Johnson's study showed that eggs laid under the calyx of lemons were susceptible to irradiation levels as low as 100 Gy, with no eggs hatching at levels above 175 Gy. This irradiation dose was comparable to other species of coleopteran reported in the literature. A detailed study is, therefore, recommended to identify an effective minimum absorbed dose for FRW in an Australian context.

### 3.4.2. Serpentine Leaf Miner

Another insect pest of concern is Serpentine leaf miner *Liriomyza huidobrensis* (Diptera: Agromyzidae) (Blanchard) detected on vegetable crops in the Sydney Basin in October 2020 [59] and American Serpentine leaf miner *Liriomyza trifolii* (Burgess) reported in western Australia by DPIRD in July 2021 [60]. Both are polyphagous, feed on most vegetable and legume crops as well as ornamental plants and pose a serious threat to Australian agriculture and horticultural industries. The infestation in NSW was significant enough that NSW DPI transitioned to management over eradication, and the pest is now considered established in both NSW and Queensland. To manage Serpentine leaf miners, integrated pest management (IPM) techniques are being used that involve a combination of biological control, modification of cultural practices and monitoring of pests and insecticides.

The previous overreliance on insecticide use has led to a rise in resistance among some of the most damaging insect pest species. This leads to increased levels of plant damage under heavy pest pressure in many countries [60]. Hence, the development of

phytosanitary irradiation dose could be an alternative control technique to manage and provide quarantine security. Some studies on phytosanitary irradiation dose development have already been performed on agromyzid leafminers. For instance, phytosanitary irradiation treatment against three species of agromyzid leafminers (Diptera: Agromyzidae), *Liriomyza sativa* (Blanchard), *L. trifolii* (Burgess) and *L. huidobrensis* (Blanchard) showed that irradiation of late-stage pupae at 150 Gy and 180 Gy prevented the formation of leaf mines by F1 offspring. Confirmatory testing on all three species using >30,000 late-stage pupae suggests that 150 Gy is sufficient to prevent the formation of leaf mines completely and, thus, can be used as a phytosanitary dose [61]. Hallman et al. [62] suggested that a generic dose for agromyzids and other insects found on cut flowers could be 250 Gy.

### 3.4.3. Vineyard Snail

Vineyard snail or common white snail *Cernuella virgata* (Da Costa) is an invasive species and an agricultural pest in many parts of southern Australia. Missed timing for treatment sprays increases the probability that snail populations will be detected at threshold levels in high-value commodities such as table grapes. Snails and slugs are cold-resistant and, thus, relatively hard to treat with standard phytosanitary cold treatment. Currently, there is no minimum absorbed dose and efficacy data for this pest. However, irradiation has the potential to treat snails and slug pests.

Little research exists on the radiosensitivity of terrestrial herbivorous gastropods, although several studies have been carried out on the medically important aquatic snail *Biomphalaria glabrata* (Say) (Planorbinae) [63]. Studies with the orchid snail *Zonitoides arboreus* (Say) (Stylommatophora: Gastrodontidae) showed reproduction could be prevented by irradiation of the pest at 70 Gy [64]. Another terrestrial species of citrus tree defoliator, the brown garden snail, *Cornu aspersum* (Müller), became sterile after irradiation with ≥75 Gy [63]. A dose of >150 Gy prevented the establishment of viable populations of semi-slug *Parmarion martensi* Simroth (Stylommatophora: Ariophantidae), a pest of sweet potato and other fruits and vegetables in Hawaii [65]. These studies suggest prevention and control of snail reproduction is possible with relatively low doses. Follett et al. [65] suggested a minimum absorbed dose of 150 Gy may be effective against many slug and snail pests.

### 3.5. Lowering of the Minimum Absorbed Dose

Although 400 Gy has been recommended to treat broad pest insect groups, many insects from the orders Diptera, Coleoptera and some Lepidopterans, as well as insects from sub-order Homoptera (aphids), do become sterile and stop further development at a lower dose [66]. In general, most insect, mite, and tick families require a sterilizing dose of <200 Gy, which could serve as a phytosanitary treatment [67]. USDA APHIS has already approved a reduction in the minimum absorbed dose accepted for specific pests such as Queensland fruit fly 100 Gy, codling moth 200 Gy, oriental fruit moth 200 Gy and mango seed weevil 300 Gy [68] (Table 2).

The following case studies from Hawaii are provided as examples where research has led to a reduction in the minimum absorbed dose accepted by the USDA-APHIS:

1. A dose of 400 Gy was accepted for white peach scale control in some papayas, but a dose of 600–650 Gy was delivered to ensure 400 Gy was absorbed. Therefore, a large-scale study was conducted to test if a lower dose could give quarantine security. Results showed that 150 Gy would be effective for quarantine control of the white peach scale (resulting in commercial doses up to 250–300 Gy) and were adequate for phytosanitary disinfestation [69].
2. A dose of 400 Gy was accepted for Hawaii sweet potato [70] until research later demonstrated that 150 Gy was also sufficient to ensure quarantine security against vine borer and sweet weevil [71]. Hawaii has been using 150 Gy for sweet potato exports to the continental U.S. since this research was published.

These case studies indicate that the accepted minimum absorbed dose for a pest insect/group may be higher than is required for effective treatment. Further large-scale confirmatory tests (as outlined in ISPM 18) could be conducted to support the use of a lower minimum absorbed dose [8]. A review of the minimum absorbed dose used within Australia for priority insect pests/groups may be warranted, where a reduction in treatment costs will benefit the Australian horticulture sector.

*3.6. Differences in Australian and Internationally Accepted Irradiation Doses for Pests of Quarantine Concern*

In Australia, the minimum absorbed dose of 150 Gy is accepted for all tephritid fruit flies, 300 Gy for mango seed weevil and 400 Gy for all other insect pests. Nevertheless, differences exist in the minimum absorbed dose required for insect pests between the scientific literature and domestic and international phytosanitary protocols (Table 2). Even then, the minimum absorbed dose specified by the research may not be accepted, and a higher rate may be employed to provide an added safety margin.

Using the fruit fly as an example, most tephritid fruit flies do not require 150 Gy for disinfestation, and lower doses of 70 Gy have already been accepted by the USDA APHIS for certain species. Native fruit flies (e.g., Queensland fruit fly and Jarvis fly) are a major concern for Australian exports, and large-scale confirmatory tests were performed on the Queensland fruit fly and Jarvis fly. Data showed that 150 Gy is not required to achieve efficacy; lower doses of 75 Gy for Queensland fruit fly [29,30] and 100 Gy for Jarvis fly [72] are sufficient for treatment (Table 2). Nevertheless, Australia accepts a minimum absorbed dose of 150 Gy. For Australia, the reasons for accepting 150 Gy might include the following (but not limited to):

- To follow FSANZ [21] requirements in Section 4 where it is specified that 'fresh fruit and fresh vegetables may be irradiated for the purpose of pest disinfestation for a phytosanitary objective if the absorbed dose is: (a) no lower than 150 Gy; and (b) no higher than 1 kGy'.
- To avoid the risk of uncertainty of disinfestation from other tephritid fruit flies, such as island flies for which no phytosanitary irradiation efficacy data are recorded (although island fly is mostly found in citrus fruits).
- To meet the phytosanitary protocols agreed with international trading partners of 150 Gy for specified commodities, such as the case for New Zealand for tephritid fruit fly.

Similar considerations will be at the forefront of all NPPOs when negotiating and agreeing on protocols for phytosanitary irradiation.

## 4. Conclusions

Most major insect groups of quarantine concern for Australia have irradiation dose and efficacy data available to facilitate effective trade using phytosanitary irradiation. Nevertheless, some gaps exist in the minimum absorbed dose and efficacy data for plant pests that impact market access and growth of Australian horticulture exports. Filling the gap for insect pests such as Fuller's rose weevil, Serpentine leaf miner and Vineyard snail would be of particular benefit to the Australian citrus, vegetable and table grape industries.

The minimum absorbed dose accepted for the phytosanitary treatment of insect pests may also differ between Australia and its international trading partners. Although the minimum absorbed dose can be reduced for some species/insect groups based on large-scale confirmatory research, phytosanitary protocols agreed upon with international trading partners often result in higher dosage rates being specified in phytosanitary protocols. However, it is generally accepted that 150 Gy is effective as a treatment for all tephritid fruit flies, and 400 Gy is effective for all insect pests in host fruits and vegetables.

The generic 400 Gy treatment suggested by the USDA APHIS in 2006 could be beneficial from an international trade perspective if endorsed as an ISPM. It provides for a single effective dose, can be used on multiple commodities, is cost-effective, simplifies quarantine

inspection and certification and reduces the need for remedial treatment scenarios. Therefore, IAEA started a new coordinated research project (CRP D61026) in 2022; research will be conducted with major pest groups to confirm generic doses under this project. Steritech Australia supports the establishment of a generic dose with the most important quarantine pests treated at either 150 Gy or 400 Gy for domestic and export purposes.

Likewise, identifying the minimum effective dose for an insect pest can also be important for specific commodities and markets. A reduction in treatment time can deliver cost savings and limit damage to horticultural products that are sensitive to irradiation, for example, treating lychee for acari mites.

To protect and grow export markets for Australian horticulture commodities, research should be conducted for specific pests where no minimum absorbed dose and efficacy data are available.

Based on this literature review, the following recommendations for further research are as follows:

1.  Develop phytosanitary irradiation minimum absorbed dose and efficacy data for vineyard snails.
2.  Develop phytosanitary irradiation minimum absorbed dose and efficacy data for leaf miners (particularly Serpentine leaf miners).
3.  Develop phytosanitary dose and efficacy data for Fuller's rose weevil.

Additional trade benefits may also be achieved from research that will support international efforts to establish 400 Gy as a generic treatment for insect pests at the international level and to lower the accepted minimum absorbed dose with our trading partners to reduce the cost of treatment for priority pests and commodities.

**Table 1.** Priority commodities that have high demand in international market and associated pests of quarantine concern for Australian horticultural industry.

| Priority Commodities | Export Value (Year) | Prospective International Market | Associated Important Pests |
| --- | --- | --- | --- |
| **Table grapes** | $555 M (2018–2019) $623 M (2019–2020) | Vietnam, New Zealand, China, Indonesia, Japan, the Philippines | Grape leaf rust mite, false red mite, yellow peach moth, false codling moth, vineyard snails, fruit flies, beetle, western flower thrips, spider |
| **Cherry** | $79.5 M (2018–2019) | Vietnam, Indonesia, China, Korea | Spider mite, oriental fruit moth, green fruit worm, leaf roller, fruit flies, black cherry aphid, black peach aphid, thrips |
| **Lychee** | $7.4 M (2018–2019) | United States, China, USA, New Zealand and Canada | Mites, moth, fruit flies, beetles |
| **Stone/summer fruit (apricots, nectarines, peaches and plums)** | $89 M (2018–2019) | China, United Arab Emirates, Saudi Arabia, Singapore and Malaysia | Mites, plum fruit moth, fruit flies, aphid, beetle, western flower thrips In 2020, summer fruit Australia was involved in several responses to pest incursions affecting the industry, including detections of brown marmorated stink bug, varroa mite and exotic fruit fly. |
| **Blackberry and raspberry** | 1.87 M (2018–2019) | China, Singapore, India, Indonesia, Pacific Island countries, the United Arab Emirates, Canada and Europe | Mites, moth, borer, fruit flies, vinegar flies, thrips |

**Table 1.** *Cont.*

| Priority Commodities | Export Value (Year) | Prospective International Market | Associated Important Pests |
|---|---|---|---|
| **Strawberry** | $24.4 M (2018–2019) | United Arab Emirates, New Zealand, Singapore, Thailand and China | Mites, moth, borer, fruit flies, vinegar flies, thrips |
| **Citrus: oranges, mandarins, lemons, limes and grapefruit** | $457 M (2018–2019) | China, Japan, Malaysia, Indonesia, United Arab Emirates, Singapore, the United States and Thailand. citrus industry is Australia's largest fresh fruit exporting industry by volume | citrus rust mite, citrus mite, false red mite, false codling moth, citrus fruit borer, fruit flies, scale, fuller rose weevil, citrus thrips |
| **Papaya** | Export value not availble | New Zealand | Moth, fruit flies |
| **Tomato** | Export value not availble | New Zealand, Indonesia | Tomato worm, borer, fruit flies, serpentine leaf miner |
| **Capsicum** | Export value not availble | New Zealand | Moth, serpentine leaf miner, fruit flies |
| **Melons** | Export value not availble | Indonesia | Fruit flies |
| **Persimmon** | Export value not availble | Thailand | Fruit flies |

Source: [73,74], Benjamin Reilly, Steritech, Melbourne.

**Table 2.** Minimum absorbed dose approved for insects/insect groups within Australia and the United States.

| Insect Pest | | Minimum Absorbed Dose (Gy) (Study Reference) | Minimum Absorbed Dose (Gy) Approved by USDA APHIS [1] | Minimum Absorbed Dose (Gy) Approved in Australia [2] |
|---|---|---|---|---|
| **Scientific Name** | **Common Name** | | | |
| *Anastrepha ludens* **(Loew)** | Mexican fruit fly | 70 Gy [75] | 70 | 150 |
| *Anastrepha obliqua* **(Macquart)** | West Indian fruit fly | 70 Gy [75] | 70 | 150 |
| *Anastrepha serpentina* **(Wiedemann)** | Sapote fruit fly | 100 Gy [33] | 100 | 150 |
| *Anastrepha suspensa* **(Loew)** | Caribbean fruit fly | 70 Gy [75] | 70 | 150 |
| *Bactrocera jarvisi* **(Tryon)** | Jarvis fly | 100 Gy [72] | 100 | 150 |
| *Bactrocera tryoni* **(Froggatt)** | Queensland fruit fly | 75 Gy [29,30] | 100 | 150 |
| *Fruit flies in the family Tephritidae not listed above* | | 150 Gy [9] | 150 | 150 |
| *Brevipalpus chilensis* **(Baker)** | False grape mite/false red spider mite (acari mite) | 300 Gy [76] | 300 | 400 |
| *Cryptophlebia ombrodelta* **(Lower)** | Litchi fruit moth | 250 Gy [77] | 250 | 400 |
| *Cydia pomonella* **(L.)** | Codling moth | 200 Gy [36] | 200 | 400 |
| *Grapholita molesta* **(Busck)** | Oriental fruit moth | 200 Gy [38] | 200 | 400 |
| *Omphisa anastomosalis* **(Guenee)** | Sweet potato vine borer | 150 Gy [71] | 150 | 400 |

**Table 2.** *Cont.*

| Insect Pest | | Minimum Absorbed Dose (Gy) (Study Reference) | Minimum Absorbed Dose (Gy) Approved by USDA APHIS [1] | Minimum Absorbed Dose (Gy) Approved in Australia [2] |
|---|---|---|---|---|
| Scientific Name | Common Name | | | |
| *Pseudaulacaspis pentagona* (Targioni Tozzetti) | White peach scale | 150 Gy [69] | 150 | 400 |
| *Aspidiotus destructor* (Signoret) | Coconut scale | 150 Gy [78] | 150 | 400 |
| *Sternochetus mangiferae* (F.) | Mango seed weevil | 300 Gy [31,77] | 300 | 300 |
| *Cylas formicarius elegantulus* (Summers) | Sweet potato weevil | 150 Gy [71] | 150 | 400 |
| *Euscepes postfasciatus* (Fairemaire) | West Indian sweet potato weevil | 150 Gy [71] | 150 | 400 |
| *Conotrachelus nenuphar* (Herbst) | Plum curculio | 92 Gy [79,80] | 92 | 400 |
| *Asynonychus cervinus* | Fuller's rose weevil | 174.1 Gy [57] | No information | Not selected yet |
| *Cernuella virgata* (Da Costa) | Vineyard snails | No information | No information | Not selected yet |
| *Liriomyza huidobrensis* (Blanchard) | Serpentine leaf miner | No information | No information | Not selected yet |
| *Liriomyza trifolii* (Burgess) | American serpentine leaf miner | No information | No information | Not selected yet |

Source: [1] [13,25,68], [2] Steritech personal communication, [21].

**Author Contributions:** H.A. collected information and exchange knowledge by meeting project collaborators, stakeholders from industry, government, and research organizations, and international experts in relevant field. Humayra collected information from published literature and reports, extracted information, assessed, researched and synthesized information. Based on discussion with coauthors, Humayra outlined review structure, wrote the first draft of this review, sent to coauthors for comments and incorporated all suggestions from coauthors and wrote the final version of review; N.C. participated in research associated with this paper, provided feedback and discussion on content and layout of the review paper and discussed final proofs with co-authors; P.R. critically reviewed the manuscript, discussed with co-authors; M.B. participated in discussion about the layout of review paper, critically reviewed the manuscript, added valuable comments and ensured the document had links to existing international/Australian standards and regulation. All authors have read and agreed to the published version of the manuscript.

**Funding:** Horticulture Innovation Australia: AM19002.

**Institutional Review Board Statement:** Not applicable.

**Acknowledgments:** This literature review was conducted as part of project AM19002—Building Capacity in Irradiation, funded by Horticulture Innovation Australia. Authors are indebted to Peter A Follett, Research Entomologist USDA-ARS, Daniel K. Inouye U.S. Pacific Basin Agricultural Research Center, for his generous support by providing literature and expert suggestions regarding phytosanitary irradiation technique and current research status in the world. Benjamin Reilly, Fresh Produce Business Manager, Steritech, Victoria, also helped by providing information about t priority crop list that are important for phytosanitary irradiation in Australia and comments on the review for which authors are grateful.

**Conflicts of Interest:** The authors declare no conflict of interest.

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
