# Peer review of "An Overview of Phytosanitary Irradiation Requirements for Australian Pests of Quarantine Concern"

_agriculture, doi:10.3390/agriculture13040771_

Round 1

Reviewer 1 Report

This review provides an Australian perspective on the growing use of phytosanitary irradiation. Several Australian pests that need study are identified, and the 400 Gy generic dose approved by USDA and FSANZ is recommended to the IPPC as ways to expand and improve the practical use and acceptance of this technology. The review is clear and concise and generally well written. A few minor comments follow:

5.1 - do not start a sentence with a citation, e.g. [32]

5.2 -  program is misspelled; also, for tortricids, USDA has approved 290 Gy and the IPPC has approved 250 Gy for control of eggs and larvae.

8. - USDA misspelled

Table 1. common insects names are capitalized in several places and should not be , e.g. Beetles, Aphids, Thrips

The review is publishable with minor revisions.

Reviewer 2 Report

1.      In section 3, please introduce the IPPC PT standards on generic doses for irradiation treatment of the genus Anastrepha (PT 39) and subfamily Tortricidae (PT 40). In addition, it is best to give a brief introduction and discussion on generic dose, for example, the new CRP program D61026 Implemented by IAEA.

2.      Section 5.3 Mites is suggested placing after section 5.4 Thrips as they belong to insect;

3.      The supplementary materials of Table 1 are useful, and the volume of irradiated fruits are essential and please add them if possible.  

Reviewer 3 Report

This manuscript should introduce the cost information of irradiation.

Based on the ISPM 5, it is better to say "pest" rather than " pests and diseases".

building and safety requirements”?

"[32] treated 100000 third"?

"pro1rams [33, 34,35, 41]"?   "snail; or", "rently; there", "However; irra"   "To follow [21] requirements"

Table 1, Hongkong is part of China.
